**communications** engineering

# Benchmarking scientific machine-learning approaches for flow prediction around complex geometries
Ali Rabeh[1,3], Ethan Herron [1,3], Aditya Balu [1], Soumik Sarkar [1], Chinmay Hegde[2], Adarsh Krishnamurthy [1] ✉ & Baskar Ganapathysubramanian [1] ✉

Rapid and accurate simulations of fluid dynamics around complicated geometric bodies are critical in a variety of engineering and scientific applications. While scientific machine learning (SciML) has shown considerable promise, most studies in this field are limited to simple geometries. This paper addresses this gap by benchmarking diverse SciML models, including neural operators and vision transformer-based foundation models, for fluid flow prediction over intricate geometries. We evaluate the impact of geometric representations—Signed Distance Fields (SDF) and binary masks—on model accuracy, scalability, and generalization using a high-fidelity dataset of steady-state flow over complex geometries. We introduce a unified scoring framework that integrates metrics for global accuracy, boundary layer fidelity, and physical consistency. Our findings reveal that newer foundation models significantly outperform neural operators, particularly in data-limited scenarios. In addition, binary mask representation enhances the performance of vision transformer models by up to 10%, while SDF representations improve neural operator performance by up to 7%. Despite these promises, all models struggle with out-of-distribution generalization, highlighting a critical challenge for future SciML applications. Our work paves the way for robust and scalable ML solutions for fluid dynamics across complex geometries.

Accurate prediction and optimization of flows around complicated geometries are essential across various engineering disciplines, since fluid interactions with intricate shapes profoundly affect real-world behavior. In fields such as aerodynamics, fluid mechanics, and bioengineering, understanding flow patterns enables the analysis of fluid behavior around structures such as aircraft wings[1], vehicle bodies[2,3], cardiovascular flows[4,5], and architectural designs[6]. Such insights are critical to advance drag reduction, lift generation, and heat transfer optimization. However, while accurate, traditional computational fluid dynamics (CFD) methods are often slow and computationally expensive, limiting their use in real-time or large-scale applications. This has led to the development of reduced-order models (ROMs) to serve as fast surrogates for expensive CFD simulations and address these limitations.

However, ROMs are generally only relevant for applications similar to the specific problems for which they were designed. This is due to their reliance on dimensionality reduction techniques to manage complex parameter spaces[7]. This lack of generalizability to new simulations or changing parameters often restricts their practical utility. However,

scientific machine learning (SciML) offers a promising alternative by leveraging data-driven techniques to model fluid dynamics with both speed and accuracy. The emergence of SciML has been driven by advancements in neural networks, which have demonstrated remarkable capabilities in learning complex, nonlinear mappings from high-dimensional data[8]. These models enable data-driven discovery, bypassing the need to solve governing equations explicitly while maintaining high levels of prediction accuracy[9]. As a result, SciML is increasingly being adopted in fields that require rapid prototyping[10], optimization[11], and quantification of uncertainty of fluid flow phenomena[12–14]. Recent research employing transformer architectures shows their efficiency in modeling fluid dynamics by uncovering key features in turbulent flow data, thereby reducing the need of large expansive datasets[15,16]. Moreover, Drikakis et al.[17] showed that these models can generalize well across a wide range of flow conditions, underscoring their ability to accurately predict flow regimes unseen during training.

Recent advances in SciML have enabled the development of models capable of approximating fluid dynamics with remarkable efficiency. These data-driven models can significantly reduce computational costs while

[1]Iowa State University, Ames, IA, USA. [2]NYU Tandon School of Engineering, New York, NY, USA. [3]These authors contributed equally: Ali Rabeh, Ethan Herron. ✉e-mail: adarsh@iastate.edu; baskarg@iastate.edu

maintaining high accuracy, making them attractive for applications where the deployment of traditional solvers is infeasible due to time constraints. However, despite these advances, SciML models have primarily been evaluated on flows over simple geometries[18–20], limiting their relevance to real-world scenarios. Applications involving complex geometries, such as urban wind flows, biomedical fluid dynamics, and turbulent flows around vehicles, remain underexplored in SciML[21]. Therefore, it is necessary to carefully benchmark SciML models on challenging datasets involving intricate boundary interactions to uncover their true potential and limitations. Our study seeks to bridge this gap by evaluating the performance of a variety of SciML models in predicting fluid-flow problems using a high-fidelity dataset of steady-state flow, governed by the Navier-Stokes equations[22] and focusing on flows around complex geometries.

Incorporating geometry into SciML frameworks has gained significant attention recently, as researchers tackle challenges such as complex boundary conditions, arbitrary domains, and out-of-distribution (OOD) generalization. A major obstacle in SciML-based flow modeling is the capacity to extrapolate to previously unseen flow conditions. Liu et al.[23] introduced a data-efficient SciML foundation model, demonstrating that large-scale pretraining of neural operators can substantially enhance their ability to generalize beyond training distributions. Furthermore, Bonfanti et al.[24] examined how hyperparameter choices impact the robust extrapolation capabilities of Physics-Informed Neural Networks (PINNs) for PDE solving in unseen regimes. In addition, current state-of-the-art SciML models continue to struggle with accurately capturing flows around complex geometries, which are characterized by steep gradients and fine-scale fluid interactions near surfaces. Collins et al.[21] proposed a neural operator surrogate to model 2D airflow over varied geometries, underscoring the sensitivity of local flow features to even minor geometric perturbations. Similarly Serrano et al.[25], introduced an operator-learning framework that leverages neural fields with latent geometry representations to handle PDEs on complex, non-convex domains. Peyvan and Kumar[26] employed a modified DeepONet architecture for hypersonic flows around parameterized geometries to predict flow over a broad range of geometric designs. Building on these advancements, the present work explores geometry representations-both binary masks and signed distance fields-specifically within SciML-based operators. This approach directly addresses the need for models that can systematically account for complex geometry and scale to unseen flow scenarios, and our work seeks to further narrow the gap in geometry-aware surrogate modeling. Despite these advances, current SciML models often fail to quantify predictive uncertainty-especially in regimes that are outside the training distribution or near boundaries influenced by complex geometries. This lack of uncertainty awareness limits their reliability in real-world applications where decisions require calibrated confidence measures. Recent efforts focused on uncertainty quantification in neural operator surrogates used methods such as variational inference and Langevin dynamics to model predictive distributions[27].

We focus on two distinct types of geometric representations: binary geometry masks and Signed Distance Fields (SDF). The geometry mask indicates whether a point lies inside or outside an object, effectively isolating the region of interest. In contrast, the SDF offers richer and smoother information by encoding the shortest distance from each point in the simulation domain to the surface of the complex geometry, and distinguishes interior from exterior points[28–30]. By systematically evaluating these representations, our goal will be to understand how effectively SciML models can capture and simulate flow features around complex geometric objects. A

A promising feature of SciML models is the ability to extrapolate to out-of-sample distributions, a crucial requirement in computational fluid dynamics (CFD)[31,32]. Unlike traditional physics solvers, designed for fixed boundary conditions and parameter settings, SciML models can generalize beyond their training distributions. However, real-world flows often involve variations in geometry, parameters, and boundary conditions, posing significant challenges for generalization[33,34]. Accurate predictions in extrapolatory regimes enable SciML models to serve as reliable alternatives to

traditional methods, improving the feasibility of modeling unseen flow scenarios[35–37]. Understanding the limits of SciML models in these regimes is critical for designing architectures and training strategies capable of achieving robust generalization, even under significant distribution shifts.

Another essential consideration is data sufficiency, which refers to the training data required for optimal model performance. Sufficient training data enables models to capture critical flow characteristics correctly[38,39]. However, obtaining large datasets is often computationally and practically challenging. This challenge is further compounded in high-fidelity CFD simulations, where generating datasets involves solving partial differential equations (PDEs) over millions of grid points, requiring significant computational resources[40]. By investigating the relationship between dataset size and model performance, this study aims to offer actionable insights into how data efficiency can be achieved without sacrificing prediction accuracy.

The FlowBench dataset[22] comprises more than 10,000 high-fidelity simulations featuring a variety of complex geometries and diverse flow conditions. This dataset is tailored for the benchmarking of scientific machine learning models, covering tasks associated with complex geometrical configurations and including both 2D and 3D simulations that capture steady and transient fluid dynamics across single and multiphysics scenarios (see Figure 1). Specifically, our study employs the 2D Lid-driven Cavity subset from FlowBench to explore the following questions:

- What is the most effective representation of complex geometries in the context of predicting fluid flow around these geometries?
- How does training dataset size correlate with performance, and what is the minimum data requirement for SciML models to predict fluid flow around complex geometries?
- How do SciML models' accuracy respond to distribution shifts between training and testing datasets, particularly in extrapolatory regimes (of both geometry and Reynolds number)?

To evaluate the performance of SciML models across these tasks, we utilize three metrics: global mean squared error (MSE), near-boundary MSE, and PDE residual[41,42]. These metrics assess the ability of the SciML model to replicate flow fields, capture boundary effects, and adhere to physical laws (PDEs). The inclusion of PDE residual as a metric is particularly appealing, as it directly measures the ability of the model to satisfy governing equations, offering insights into the physical consistency of their predictions. Since it is cognitively challenging to evaluate models across different metrics, we define a single unified score normalized to the range of 0 to 100. This score is calculated based on the logarithmic scale of MSE values, where $MSE_{max} = 1$ corresponds to a meaningless prediction (e.g., predicting zero everywhere) and $MSE_{min} = 10^{-6}$ reflects the numerical accuracy of the CFD simulations. This scoring system ensures a meaningful comparison by aligning with both the worst-case scenario (score of 0) and the expected precision of the ground truth (score of 100). Further details of the scoring scale are provided in Evaluation Metrics. Results describes the dataset, details the SciML models employed, and presents the experimental results and their analysis. Discussion summarizes the findings, identifies open questions, and discusses potential directions for future research.

## Results
### Geometric representation
We evaluate two geometric representations: SDF and binary masks. SDFs are a scalar field indicating the shortest distance from each point in the prediction domain to the object's boundary. The signed distance field represents the shortest distance from a given point in space to the surface of a geometric shape. It takes negative values inside the object, positive values outside the object, and zero values on the boundary surface. In contrast, the binary mask represents geometry as a binary field, with 0 inside the object and 1 outside, offering a more straightforward, less informative structure regarding relative distances to the boundary layer. An example of the SDF and binary mask for three sample geometries from our dataset is shown in Fig. 2. We aim to assess whether there is added value in using a continuous

**Fig. 1 | A data-driven evaluation framework for accelerating PDE solvers of fluid flow around complex geometries using scientific machine learning models.** This figure illustrates the scientific ML framework that assesses neural operators and foundation models for fluid flow solvers. Flow simulations are performed for steady-state lid-driven cavity flow across various complex geometries and Reynolds numbers. The top left subpanel shows model inputs, including a randomly selected geometry with two inputs: the Reynolds number and a representation of the geometry. It also shows model outputs---x-velocity (*u*), y-velocity (*v*), and the pressure field. The bottom subpanel presents 15 randomly chosen geometries and a Reynolds number distribution bar chart of the training dataset. The top right subpanel contrasts two representations of a sample geometry: the binary mask and the signed distance field (SDF).

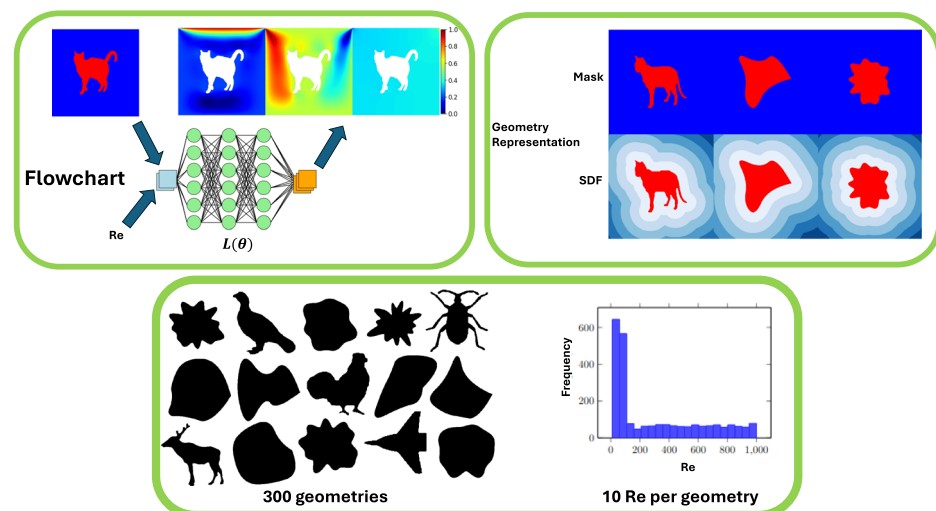

**Fig. 2 | Comparison of geometry representations for different geometries. a–c** Signed Distance Field (SDF) representations and **d–f** binary mask representations.

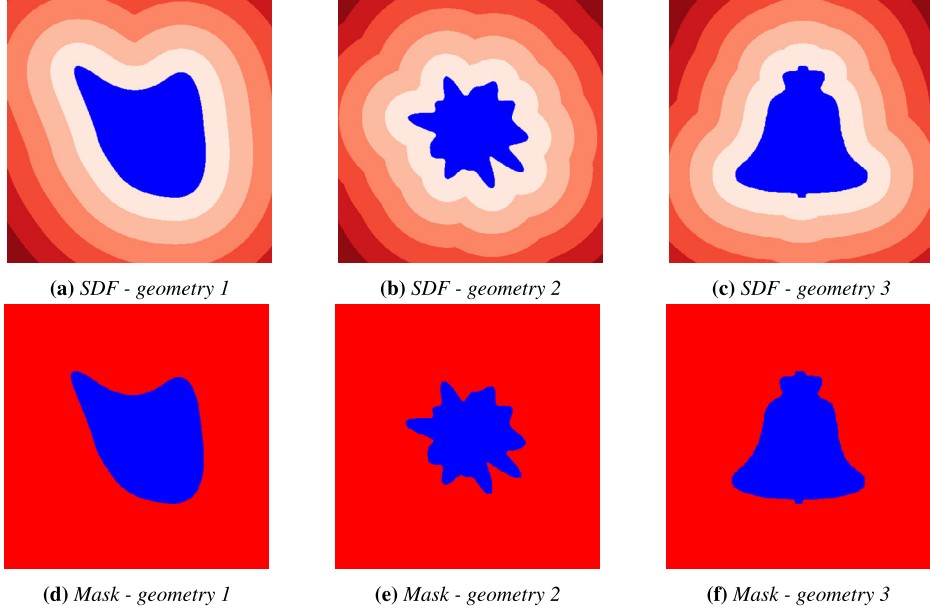

**(a)** *SDF - geometry 1*     **(b)** *SDF - geometry 2*     **(c)** *SDF - geometry 3*

**(d)** *Mask - geometry 1*     **(e)** *Mask - geometry 2*     **(f)** *Mask - geometry 3*

representation of distance from object boundary versus a simple binary mask on capturing fluid behavior around objects.

To quantitatively evaluate the impact of geometric representation on the performance of the SciML model, we utilize a unified scoring scale. This scale presents error metrics in an understandable range between 0 and 100. A score of 0 indicates the least favorable outcome, where the models predict zero across all fields, and a score of 100 corresponds to model predictions that align with the high precision of computational fluid dynamics (CFD) simulations.

We assess the effect of geometry representation on SciML prediction error. The dataset, consisting of 3000 samples, is randomly divided into an 80-20 train/test split. The test dataset, containing 600 samples, is held constant across all experiments to ensure consistent evaluation of model performance. As shown in Fig. 3, scOT-T, poseidon-T, and CNO achieve higher scores with the mask representation, while other neural operators tend to perform better with the SDF. Additionally, Tables 1, 2 show that scOT and Poseidon models outperform the other neural operators by roughly an order of magnitude. Sample field predictions and error comparisons between SDF and mask representations for the velocity in

the y-direction of a random test sample are displayed in Fig. S.1. The error plot indicates that the mask representation produces lower error for the poseidon-T and CNO models, whereas the SDF yields lower error for the geometric-DeepONet model. This difference suggests that scOT, Poseidon, and CNO models benefit from the sharpness of the binary mask, while other neural operators perform better when using the continuous boundary information provided by the SDF. This observation is not intuitive as SDF is a richer field that provides information on how close the object boundary is versus simple in/out information through the binary mask.

**Key takeaways**.

- Impact on Model Performance: Vision transformer-based models such as scOT-T and poseidon-T, along with the CNO model, exhibit improved accuracy when using binary mask representations, while other neural operators perform better using the SDF.

- Performance Comparison: scOT and Poseidon models outperform other neural operators scoring 20 points higher in performance metrics (corresponding to an order of magnitude lower MSE).

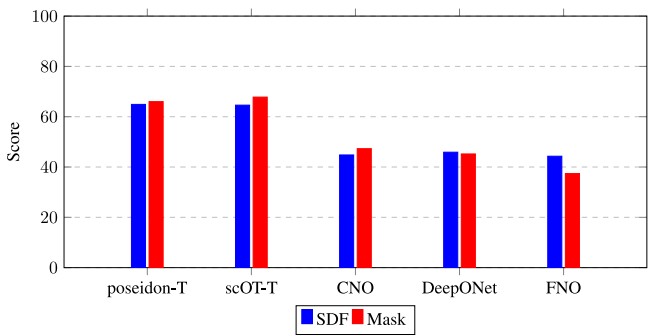

**Fig. 3 |** Comparison of score values for different models using Signed Distance Field (SDF) and binary mask representations. The scores are computed on the test dataset using a normalized error metric based on the M1 global accuracy metric (SDF≥0). The bar plot shows the score for each model, indicating the performance difference between SDF and mask representations.

### Table 1 | The score of SciML models trained on the full dataset using the signed distance field at two different difficulty levels (random and extrapolatory)

| Model | Random | | | Extrapolatory | | |
|---|---|---|---|---|---|---|
| | M1 | M2 | M3 | M1 | M2 | M3 |
| **poseidon-L** | 55.9 | 65.2 | 24.1 | 22.5 | 41.6 | 0.0 |
| **poseidon-B** | 58.7 | 69.4 | 23.6 | 26.5 | 41.2 | 28.2 |
| **poseidon-T** | **64.9** | **73.3** | 24.2 | 27.0 | **42.2** | 26.6 |
| **scOT-L** | 60.0 | 68.6 | 23.6 | 21.9 | 38.8 | 29.5 |
| **scOT-B** | 58.3 | 67.0 | 24.1 | 23.6 | 38.4 | 28.1 |
| **scOT-T** | 64.6 | 71.4 | 23.5 | 23.7 | 38.9 | 29.3 |
| **CNO** | 44.8 | 54.5 | 28.2 | 24.1 | 36.4 | 24.9 |
| **FNO** | 44.3 | 59.2 | 20.4 | 18.2 | 32.2 | 45.0 |
| **WNO** | 24.1 | 41.3 | 27.7 | 13.6 | 28.4 | 7.7 |
| **Deeponet** | 45.9 | 53.0 | **33.5** | **28.0** | 35.6 | 27.7 |
| **geometric-deeponet** | 53.0 | 59.9 | 30.2 | 25.0 | 37.4 | **30.8** |

In this table, M1 denotes the global accuracy ($SDF \geq 0$), M2 denotes the boundary layer accuracy ($0 \leq SDF \leq 0.2$), and M3 denotes the physical consistency using the $L_2$-norm of the momentum residuals. All errors are reported on the testing dataset.
**Bold values** indicate the best-performing model for that metric within each difficulty setting.

## Data sufficiency

Evaluating the impact of training dataset size is critical for understanding the practical feasibility of deploying SciML models, especially in scenarios where generating large datasets is computationally expensive or time-intensive. To investigate the role of training dataset size in the performance of SciML models, we conduct a series of experiments using subsets of the FlowBench dataset. The baseline experiment uses the full training dataset of 2,400 samples, representing the complete FlowBench dataset. Four additional experiments are performed with subsets of 1200, 800, 400, and 240 samples, while the same fixed test dataset of 600 samples is used for all experiments to calculate the error metrics. This ensures that any performance changes are solely due to variations in the training dataset size, and not from differences in test data. These experiments address key questions regarding data sufficiency: How much data is required to achieve reasonable performance? Are there general trends in data requirements across the SciML models? Do certain models demonstrate greater data efficiency than others, and does the choice of geometry representation (e.g., SDF vs. mask) influence these trends?

In Fig. 4, model performance varies substantially based on the geometry representation, with notable differences between the Signed Distance Field (SDF) and binary mask. Across all models-poseidon-T, scOT-T,

scOT-B, CNO, FNO, and Geometric-DeepONet-larger sample sizes consistently lead to progressively higher score values. However, neural operators reach an asymptotic error limit of around 800 samples in the mask representation, where additional data has minimal impact on further increasing score values. In contrast, scOT and Poseidon models maintain improvements up to 1200 samples, demonstrating their capacity to effectively utilize larger data sizes. This trend underscores the influence of a smooth geometry representation, such as SDF, in enhancing model learning and highlights the ability of scOT and Poseidon models to leverage larger sample sizes due to their larger architectures.

This pattern reveals that when trained with the simpler binary mask, neural operators reach their data utility limit at approximately 800 samples. In contrast, with the SDF representation, these models continue to improve with more data. Poseidon-T, in particular, highlights the benefits of pre-training in data-limited scenarios. When trained on fewer than 800 samples, Poseidon significantly outperforms scOT, achieving an MSE that is an order of magnitude lower as shown in Fig. 4. This performance advantage is especially notable compared to other neural operators, as poseidon-T and scOT-T achieve MSE values around $10^{-4}$ (score = 50) in data-sparse scenarios, emphasizing their efficiency.

The training dataset often does not fully represent the target distribution for which the model is designed. To address this, we assess the model's ability to extrapolate and make out-of-distribution predictions ("extrapolatory") by employing a test dataset that includes field solutions for lid-driven cavity flows with Reynolds numbers either in the top or bottom 10% of the range while restricting the training dataset to Reynolds numbers from the middle 80%. While the scOT and Poseidon models consistently outperform other neural operators across both geometry representations, their performance remains stable in the extrapolatory split regardless of dataset size. This observation suggests that in out-of-distribution scenarios, the ability to extrapolate relies more on the inherent robustness of the model architectures than on the volume of training data. Detailed results for all models at smaller dataset sizes are provided in the Data Sufficiency section in the Supplementary (Tables S.1–S.8). Additionally, sample field predictions and error comparisons for models trained on 240 versus 800 samples, specifically for y-velocity of an example sample, are shown in Fig. S.2 in the Supplementary.

### Key takeaways.

- Impact of Sample Size on Performance: Neural operators benefit from increased data sizes when using the SDF representation, showing continuous improvement, whereas their performance saturates around 800 samples with the binary mask.
- Performance in Data-Limited Scenarios: pre-trained model Poseidon-T demonstrates superior accuracy in data-limited scenarios, reaching an MSE around $10^{-4}$ (score = 50) with fewer than 800 samples.

## Extrapolation capabilities

The training dataset often fails to fully represent the target distribution for which the model is intended or may not encompass its entire range. We design two train-test splitting strategies to evaluate the model's ability to make out-of-distribution predictions. For the out-of-distribution experiment, the test dataset comprises field solutions for lid-driven cavity flows with Reynolds numbers in the top or bottom 10% of the range, while the training dataset is restricted to Reynolds numbers from the middle 80%. In contrast, the baseline experiment employs a random train-test split, ensuring that both datasets contain samples spanning the entire distribution of Reynolds numbers. The distributions of Reynolds numbers for both splitting strategies are shown in Fig. 5.

We train models for each geometric representation using both random and extrapolatory datasets. As shown in Fig. 6, notable differences exist among the models for the extrapolatory data split and substantial performance gaps between each model's random and extrapolatory splits. Specifically, models trained on the random split show stronger performance, with Poseidon and scOT achieving nearly an order of magnitude lower error

than other models. For neural operators such as FNO, DeepONet, Geometric-DeepONet, and WNO, we observe that using the SDF as a geometric representation provides marginal but consistent improvements for the extrapolatory data split. This can be attributed to the SDF's ability to encode richer geometric information, including the precise location and structure of objects within the domain, compared to the binary mask's simpler representation. These results highlight the ongoing challenge of accurately

**Table 2 | The score of SciML models trained on the full dataset using the binary mask at two different difficulty levels (random and extrapolatory)**

| Model | Random | | | Extrapolatory | | |
|---|---|---|---|---|---|---|
| | M1 | M2 | M3 | M1 | M2 | M3 |
| poseidon-L | 65.6 | 74.7 | 24.1 | 20.8 | 36.3 | 26.4 |
| poseidon-B | 63.0 | 73.1 | 23.7 | 25.1 | **37.9** | 28.4 |
| poseidon-T | 66.0 | **76.1** | 23.5 | 23.4 | 37.1 | 29.8 |
| scOT-L | 67.1 | 75.6 | 24.0 | 20.1 | 34.7 | 32.1 |
| scOT-B | 62.0 | 71.8 | 24.2 | 20.4 | 33.8 | 31.5 |
| scOT-T | **67.8** | 75.1 | 24.3 | 23.1 | 36.8 | 31.5 |
| CNO | 47.3 | 59.8 | 27.1 | **26.5** | **37.9** | 28.4 |
| FNO | 37.4 | 58.5 | 25.6 | 15.6 | 29.6 | 39.2 |
| WNO | 24.8 | 41.2 | **29.5** | 11.6 | 25.0 | **40.9** |
| Deeponet | 45.2 | 53.6 | 28.7 | 20.7 | 32.4 | 34.5 |
| geometric-deeponet | 47.4 | 54.8 | 28.6 | 20.9 | 34.5 | 35.1 |

In this table, M1 denotes the global accuracy ($SDF \geq 0$), M2 denotes the boundary layer accuracy ($0 \leq SDF \leq 0.2$), and M3 denotes the physical consistency using the $L_2$-norm of the momentum residuals. All errors are reported on the testing dataset.
**Bold values** indicate the best-performing model for that metric within each difficulty setting.

extrapolating to out-of-distribution complex fluid dynamics simulations. To further illustrate this, we provide sample field predictions and error comparisons between models trained on the random and extrapolatory splits, focusing on y-velocity for an example sample, as shown in Fig. S.3 in the Supplementary. While our results indicate that transformer-based models such as Poseidon and scOT achieve lower errors for in-distribution flow conditions, their performance in extrapolatory regimes remains suboptimal. This suggests that state-of-the-art architectures struggle to generalize to unseen geometries and Reynolds number cases, highlighting a critical gap in current SciML modeling abilities.

**Key takeaways.**

- Extrapolation Challenges: Testing on extreme Reynolds numbers (top and bottom 10%) reveals that all models substantially underperform when generalizing to out of distribution predictions, with only minimal differences in performance observed between them. This indicates a challenge in accurately predicting flow behavior in extrapolatory regimes, regardless of the model architecture or geometric representation.

**Performance metrics**

We evaluate model performance using three metrics: global accuracy ($M1$), boundary layer accuracy ($M2$), and physical consistency ($M3$). Global accuracy ($M1$) measures overall prediction accuracy across the domain, excluding the geometry. Boundary layer accuracy ($M2$) focuses on errors within the boundary layer (SDF between 0 and 0.2), highlighting precision near surfaces. Physical (PDE) consistency ($M3$) assesses adherence to governing laws by evaluating momentum residuals, ensuring the physical plausibility of predictions. Notably, our results consistently show that boundary layer MSE errors ($M2$) are lower than global MSE errors ($M1$) across all SciML models, as the velocity values are close to zero near the geometry, resulting in a lower absolute MSE within the boundary layer. This observation is consistent across all results in Table 1, Table 2, and

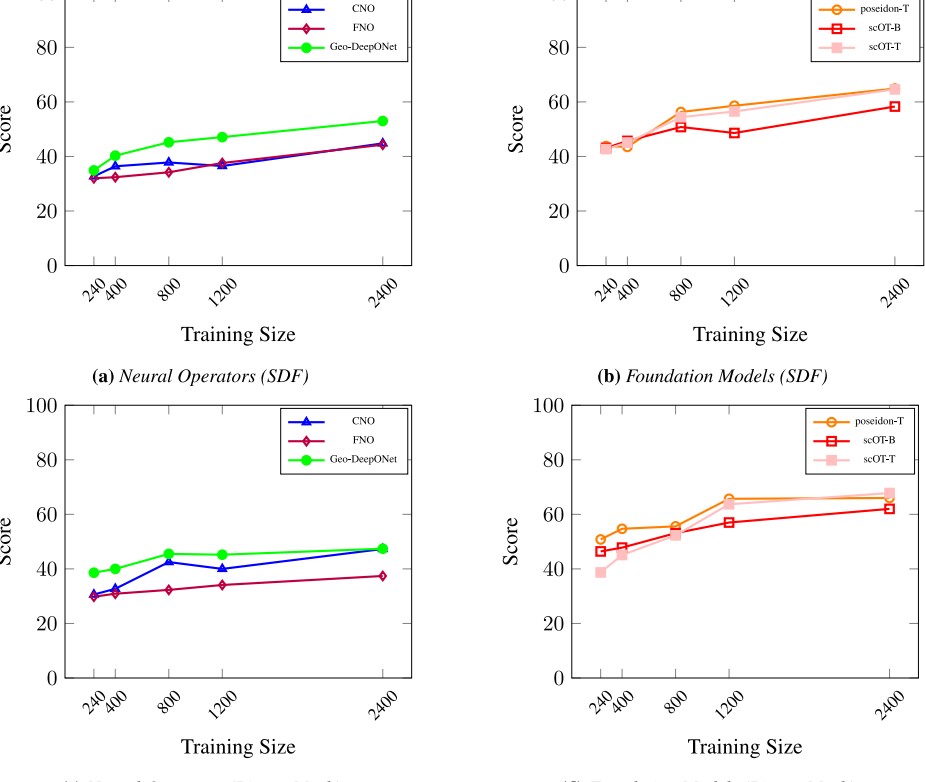

**Fig. 4 | Comparison of score values vs. sample size for different scientific machine learning models using SDF and binary mask representations.** **a** Neural operators with SDF representation (CNO, FNO, Geo-DeepONet). **b** Foundation models with SDF representation (scOT-B, scOT-T, poseidon-T). **c** Neural operators with binary mask representation. **d** Foundation models with binary mask representation. Scores are computed on the test dataset using a normalized error metric based on the M1 global accuracy metric ($SDF \geq 0$).

**(a)** *Neural Operators (SDF)* **(b)** *Foundation Models (SDF)*

**(c)** *Neural Operators (Binary Mask)* **(d)** *Foundation Models (Binary Mask)*

**Fig. 5 | Histogram of Reynolds numbers for train and test splits. a** Training data with random split. **b** Training data with extrapolatory split. **c** Test data with random split. **d** Test data with extrapolatory split.

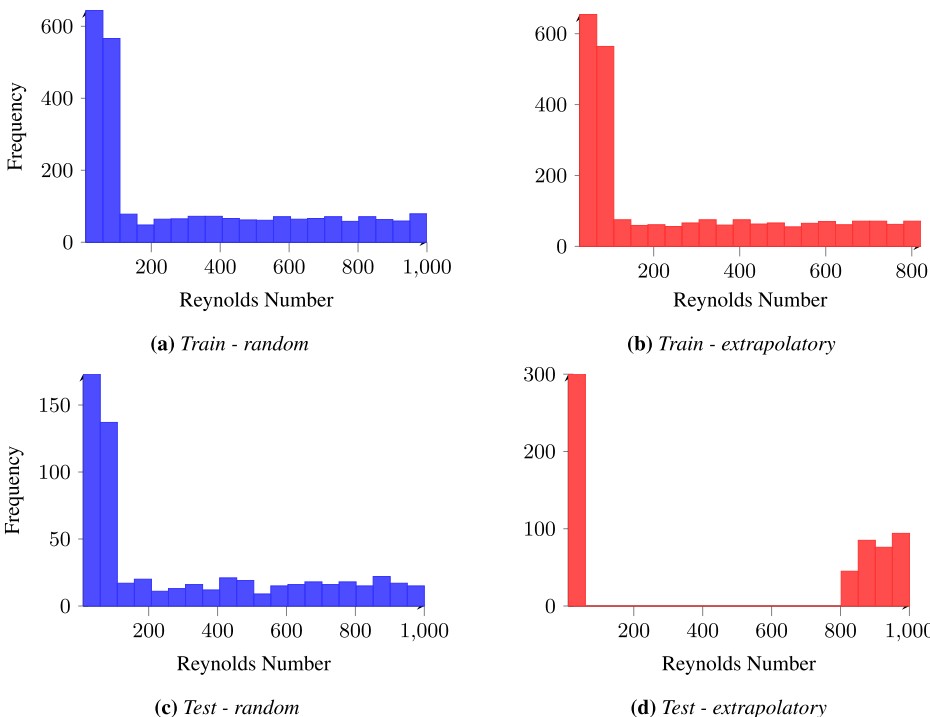

Tables S.1–S.8. Performance in the boundary layer is critical for downstream tasks such as calculating the coefficients of lift and drag using SciML predictions.

The $M3$ metric, defined as the $L_2$ norm of the momentum residuals $\left(\sqrt{r_x^2 + r_y^2}\right)$, evaluates the models' ability to satisfy underlying physical laws (PDEs) in fluid dynamics simulations. Analysis of the $M3$ metric reveals that DeepONet consistently achieve the lowest $M3$ error across all dataset configurations, as shown in Table 1, Table 2, and Tables S.1 to S.8. Vision transformer-based foundation models leverage their image-focused architecture for efficient feature extraction. However, they may struggle to capture fine-scale details and global dependencies in high-resolution scenarios, which can lead to less smooth outputs. In contrast, neural operators such as DeepONet are designed to learn mappings between function

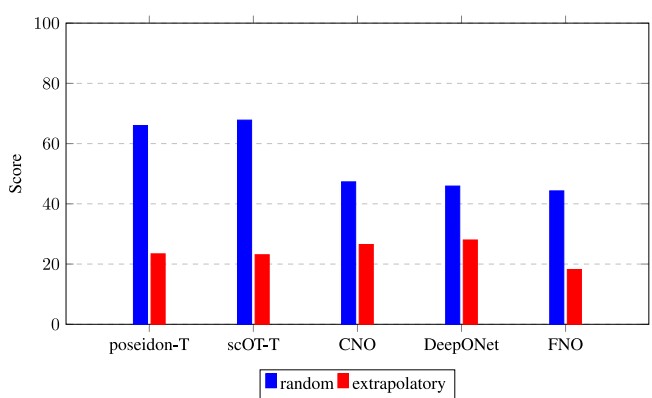

**Fig. 6 | Comparison of score values for different models using random and extrapolatory test/train splits.** The scores are computed on the test dataset using a normalized error metric based on the M1 gloabal accuracy metric (SDF≥0). The bar plot shows the score for each model, indicating the difference in performance between random and extrapolatory data splits. Poseidon-T, scOT-T, and CNO use a binary mask representation of the geometry, while DeepONet and FNO use a Signed Distance Field (SDF).

spaces allowing them to produce smoother and continuous predictions. Although neural operators architecture is not specifically tailored only for physics problems, the smoothness of the learned mappings is suitable for problems governed by partial differential equations. Consequently, DeepONet exhibits lower errors on the $M3$ physical consistency metric (momentum residuals), which is further explained in the Residual Calculation section in the Supplementary.

As illustrated in Fig. S.4, which compares residuals in the $x$ and $y$ directions for mask and SDF representations in Poseidon-T, CNO, and DeepONet, there is no significant difference in residual values between the SDF and mask representations. However, DeepONet outperforms the other models in the $M3$ metric, achieving the lowest residuals for the sample shown. In particular, the element-wise momentum residual of DeepONet is only non-zero near the top boundary and the surface of the geometry, where the velocity and pressure gradients are the highest. Conversely, Poseidon-T, and to a lesser extent, CNO, exhibit relatively high residuals throughout the domain. This finding suggests that, although it does not have the lowest MSE, DeepONet achieves a more accurate approximation of the solution gradient (see Fig. S.5), effectively satisfying the underlying PDE behavior more robustly. Additionally, residual values are generally lower in the extrapolatory split than in the random split, likely due to the higher proportion of samples with low Reynolds numbers in the extrapolatory split, simplifying the enforcement of PDE constraints for SciML models.

**Key takeaways.**
- Boundary vs. Global Accuracy ($M1$, $M2$): Boundary layer MSE ($M2$) is consistently lower than global MSE ($M1$), as the reduced velocity near the geometry leads to naturally lower absolute errors. This indicates that models effectively learn the near-zero velocity conditions around the geometry.
- PDE Consistency ($M3$): DeepONet achieves the lowest $M3$ error by leveraging its basis-function based architecture to model continuous fields. This enables DeepONet to capture smoother solutions and thus better satisfy governing equations compared to vision transformer-based models, which excel in feature extraction.

**Table 3 | Performance metrics of SciML models**

| Model | Model Size (Million) | Training Time (hr) | Inference Time (s) |
|---|---|---|---|
| Deeponet | **0.9** | 5.86 | **0.61** |
| Geometric-deeponet | 2.1 | 6.67 | 1.04 |
| FNO | 10.9 | **2.29** | 11.46 |
| CNO | 11.7 | 6.04 | 12.02 |
| scOT-T | 20.7 | 6.48 | 49.25 |
| scOT-B | 157 | 12.93 | 67.48 |
| scOT-L | 628 | 22.50 | 81.30 |
| poseidon-T | 20.7 | 6.25 | 38.81 |
| poseidon-B | 157 | 13.25 | 65.93 |
| poseidon-L | 628 | 21.94 | 80.41 |

Model size (in million parameters), training time (in hours) on the full dataset of 2400 samples, and inference time (in seconds) evaluated on the randomly split test dataset containing 600 samples.

## Computational performance and parameter analysis

The numerical efficiency of the different SciML models, including model size, training time, and inference time, is presented in Table 3. The scOT and Poseidon models exhibit similar values across these metrics due to their shared architecture. Among all models, FNO stands out as the fastest model to train, completing training in just 2.29 hours, due to its relatively simple architecture and a smaller number of parameters compared to other large models. DeepONet and geometric-DeepONet are the fastest in inference time, with DeepONet requiring less than a second per sample due to its small model size. On the other hand, the larger models, such as WNO and the base and large versions of scOT and poseidon, take significantly longer to train due to their substantial number of parameters.

Ideally, one would want to evaluate the performance of these models with same parameter count. However, training and evaluating the performance of models with equal parameter counts across these diverse architectures—ranging from neural operators such as FNO, CNO, and DeepONet to vision transformers such as scOT and Poseidon—is very challenging in practice. These architectures have fundamentally distinct structural paradigms and inductive biases; for example, FNO learns global mappings in the Fourier domain while scOT/Poseidon utilizes attention mechanisms that naturally require higher parameter counts for query-key-value projections. Forcing equal parameter count would require non-standard configurations that could impair each model's ability to learn relevant features (Recent theoretical work[43] also suggests this, i.e. architectural choices operate on different parameter scales). Thus, comparing well-tuned instances of each architecture, each operating on its design-optimal parameter scale, provides a more realistic assessment of their strengths and weaknesses. This approach, consistent with standard benchmarking practice[44], highlights the trade-offs between computational cost and predictive performance in SciML applications.

Figure 7 shows that for the global accuracy metric $M1$, neural operators such as DeepONet, FNO, and CNO, which have relatively small model sizes (under 15 million parameters), perform poorly compared to the much larger vision transformer-based models such as scOT and Poseidon. In contrast, the residual consistency metric $M3$ shows the opposite pattern: smaller models, particularly DeepONet and Geometric-DeepONet, outperform their larger counterparts. These models yield lower residual errors, suggesting a better ability to capture the underlying physical operator. This result underscores the efficiency of compact neural operators and indicates the need for more data for larger models (Scot and Poseidon) to learn the underlying physics.

## Key takeaways.

- Computational efficiency vs. accuracy: Larger foundation models achieve higher global accuracy ($M1$), but require significantly more training time. In contrast, smaller neural operator models are faster to train and deploy, and larger models in terms of physical consistency ($M3$).
- Compact models generalize better physically: Smaller and more interpretable architectures appear to be better able to learn physically consistent operators from limited data.

## Discussion

This study addresses the challenge of simulating fluid dynamics around complex geometries. Traditional simulation based methods, while accurate, are computationally expensive, prompting the exploration of scientific machine learning (SciML) for faster, scalable solutions. By benchmarking neural operators and vision transformer-based models, this work evaluates geometric representations, data efficiency, and out-of-distribution generalization, offering insights into SciML's capabilities and limitations for complex flow prediction. Key observations from our experiments include:

### Geometric representation

The choice of geometric representation influences model performance. The SDF representation generally improves the performance of neural operators such as geometric-DeepONet by offering detailed boundary information and a continuous field representation. In contrast, Poseidon, scOT, and CNO models tend to perform better with the binary mask, leveraging its sharp contrasts to focus on key geometric features.

### Data sufficiency

The impact of data sufficiency depends on geometric representation and model type. For the mask representation, neural operators saturate at a score of around 40. With only 300 training samples, Poseidon and scOT models using the mask representation achieve a score of around 50 (MSE $10^{-4}$), highlighting their efficiency in low-data scenarios. Interestingly, in the extrapolatory split, MSE marginally changes across dataset sizes, suggesting that training data size has minimal impact on performance in out-of-distribution predictions.

### Data scaling

While our overall trend shows that performance improves with increased training data, minor fluctuations are expected in such complex settings and data. In particular, the CNO model exhibited a slight dip in test performance between 800 and 1200 samples using the binary mask representation. We maintained fixed hyperparameters (tuned on a dedicated validation set) across all training sizes to isolate the effect of data scaling, and such stochastic variations-due to factors like random weight initialization and fixed learning schedules-do not contradict the overall trend of improved performance or saturation with larger training datasets.

### Model comparison

Across all experiments, the scOT and Poseidon models consistently deliver the best performance, often by a significant margin. This superiority stems from the scOT architecture's resemblance to vision transformer (ViT) models, which have demonstrated remarkable success across various image-based tasks. Our findings indicate that *scalable Operator Transformers* outperform other SciML models in modeling fluid dynamics over complex geometries. Notably, scOT and its pretrained foundation model counterpart, Poseidon, have minimal performance differences. Investigating the impact-or lack thereof-of pretraining in foundation SciML models remains an avenue for future research. Among neural operators, results vary across metrics. DeepONet and FNO excel on the $M3$ metric, reflecting superior physical consistency, while CNO achieves better results on the $M1$ and $M2$ metric, highlighting precision in global/boundary layer field predictions.

### Open questions

While our study provides insights, several open questions remain: (a) *Incorporating Physics losses*: Incorporating physics-based losses, such as the

**Fig. 7 | Effect of model size on performance.**
**a** Global accuracy metric M1 and **b** physical consistency metric M3, each plotted against total parameter count on a log scale.

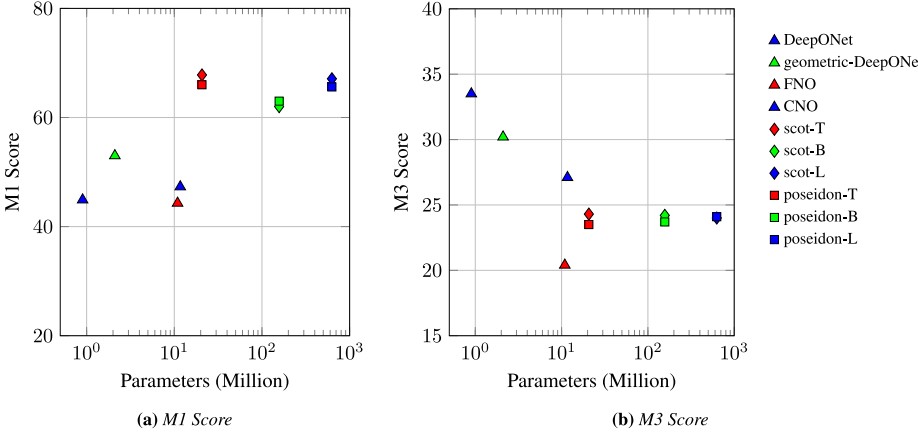

**(a)** *M1 Score* **(b)** *M3 Score*

Navier-Stokes momentum or continuity equations, could help penalize high-gradient regions and improve overall model accuracy. (b) *Robust Foundation Models*: Throughout all experiments, scOT models and their Poseidon counterparts demonstrated comparable performance, with scOT outperforming Poseidon in several scenarios. This observation highlights the need to further investigate the underlying factors contributing to this discrepancy, offering an opportunity to refine and optimize foundation models for fine-tuning on fluid dynamics across complex geometries. (c) *Multiphysics Simulations*: The combination of complex geometries with multiphysics flow phenomena (like buoyancy-driven flows (thermal or concentration), electrokinetic flows, or multi-phase flows) is the next frontier. Our results suggest opportunities and promise for these SciML models, suggesting the need for the creation and curation of such datasets. (d) *Improving Out-of-Distribution Generalization*: Although our models perform well on in-distribution data, they consistently fall short when faced with unseen flow conditions. Enhancing the capacity to generalize beyond the training regime is critical for realistic applications. Future research should explore incorporating physics-based constraints, such as velocity solenoidality, to improve robustness in conditions beyond the encountered during training. However, embedding physics-based constraints as regularizers is a challenging task that may require fundamental modifications to the design of neural operators. Current state-of-the-art neural operators are purely data-driven and lack inherent mechanisms to enforce physics invariants. Moreover, balancing the influence of physics-based regularization with the standard data-driven loss is highly nontrivial, as improper weighting can destabilize the training process and compromise data fitting.

## Methods
### Training data
In this paper, we utilize the FlowBench dataset publicly available on Hugging Face at https://huggingface.co/datasets/BGLab/FlowBench/tree/main[22]. FlowBench is specifically designed for complex fluid dynamics and heat transfer problems, providing high-fidelity simulations for challenging scientific ML models. The dataset includes 300 diverse parametric and non-parametric geometries that capture intricate flow patterns and transitions around complex shapes. Each geometry is paired with 10 Reynolds numbers, randomly selected between 10 and 1000, to allow for studying flow behavior under varying conditions. Data generation uses a validated framework based on the Navier-Stokes equations using the shifted boundary method to impose boundary conditions on surrogate boundaries[28,45].

While most research has focused on flow around simple geometries, FlowBench targets more complex shapes, categorized into three distinct types. The first category consists of parametric shapes created using Non-Uniform Rational B-Splines (NURBS) curves, commonly used in computer graphics and CAD for flexible modeling of complex shapes[46]. These shapes are generated by varying control points to produce diverse, smooth curves. The second category features parametric shapes defined by spherical

harmonics, allowing for the creation of smooth, radial geometries[47]. The third category includes non-parametric shapes sourced from the SkelNetOn dataset, which provides grayscale images of various objects[48,49]. These categories offer a broad range of shapes for evaluating model performance on fluid dynamics tasks.

This study focuses on the 2D lid-driven cavity flow (LDC) problem using the Navier-Stokes equations. The LDC setup, a canonical problem in fluid dynamics, features a square cavity with three stationary walls and one moving lid, generating complex internal flow structures such as vortices, recirculation zones, and transitions from laminar to turbulent regimes[50,51]. The FlowBench dataset is formatted as compressed numpy (.npz) files, with input fields including Reynolds numbers, geometry masks, and Signed Distance Fields (SDF) to provide geometric and physical data. The 2D LDC dataset in FlowBench contains 300 geometries with 10 simulations per geometry totaling 3000 samples, with each sample providing output fields such as velocity components ($u$ and $v$) and pressure at $512 \times 512$ resolution. The Reynolds numbers are randomly selected for each geometry, with five values ranging from 10 to 100 and another five from 100 to 1000. By varying Reynolds numbers across different geometries, FlowBench offers an extensive platform for benchmarking scientific machine-learning models on flow prediction across diverse flow regimes.

### SciML models
We evaluate 11 different SciML models; six scalable Operator Transformers with different sizes (with/without pretraining), Convolutional Neural Operators, Fourier Neural Operator, DeepONet, Geometric-DeepONet, and the Wavelet Neural Operator. For clarity, we will refer to the spectral-based neural operators, including the Convolution Neural Operator, Fourier Neural Operator, DeepONet, Geometric-DeepONet, and the Wavelet Neural Operator as *neural operators*.

The *Scalable Operator Transformers (scOT)*, the base model for the Poseidon foundation models[52], is a hierarchical, multiscale vision transformer combining Swin Transformer V2[53,54] blocks with ConvNeXt[55] residual blocks in a U-Net encoder-decoder style. The Poseidon models are scOT models pretrained on a dataset of compressible Euler and incompressible Navier-Stokes equations, available in three sizes: tiny (20M parameters), base (160M parameters), and large (630M parameters). We also include randomly initialized versions, totaling six scOT-based models. For clarity, we refer to these as scOT for the randomly initialized models and Poseidon for the pretrained models, along with their model size designation, e.g., scOT-T for the tiny randomly initialized configuration or Poseidon-B for the pretrained base configuration.

The *Fourier Neural Operators (FNO)*[56] leverages Fourier transforms to efficiently capture global interactions within the data, achieving state-of-the-art performance in problems with long-range dependencies, such as turbulence and complex flow dynamics. *Convolutional Neural Operators (CNO)*[57], which extends standard convolutional architectures to learn operator mappings through a U-shaped architecture. CNO employs

convolutional kernels to capture localized phenomena while preserving the function's continuous-discrete equivalence, making it effective for modeling fluid flow problems. *Wavelet Neural Operator (WNO)*[58], employs wavelet transformations to capture multiscale interactions by decomposing input functions into frequency bands, making it effective for handling problems with both local and global variations.

The *DeepONet*[59] employs a dual-network architecture, consisting of branch and trunk networks, to separately model input functions. Inspired by the universal approximation theorem for arbitrary continuous functions[60], this architecture enables DeepONet to flexibly represent nonlinear operator relationships. *Geometric-DeepONet*[61] builds on the standard DeepONet by integrating information about geometric representations into the trunk network, enhancing accuracy in scenarios where geometric shapes are included in the domain.

## Evaluation metrics

We evaluate the performance of a suite of SciML models in simulating the complex fluid dynamics of the lid-driven cavity flow dataset from Flow-Bench. Our experimental framework is structured around three core aspects: *geometric representation*, *data sufficiency*, and *ability to extrapolate*. All models were tuned and trained on a single A100 80GB GPU using the Adam optimizer for 200 epochs. The list of hyperparameters used for each model is provided in the Supplementary. The training and validation loss for four representative models is shown in Fig. S.6 in the Supplementary.

To thoroughly assess the performance of trained models, we introduce a hierarchical framework of evaluation metrics ($M1$, $M2$, $M3$), each designed to assess distinct aspects of model accuracy.

- $M1$: *Global accuracy*: This metric evaluates the overall prediction accuracy across the entire domain (excluding geometry). It assesses how closely the model's predicted velocity and pressure fields align with actual values, providing a comprehensive view of its general performance across the domain.
- $M2$: *Boundary layer accuracy*: This metric measures errors in the boundary layer, a critical region surrounding the object. Defined by the Signed Distance Field in the range $0 \leq SDF \leq 0.2$, this metric focuses on the model's ability to capture near-surface dynamics, essential for high-fidelity applications such as 3D manufacturing, design optimization, and control. By evaluating errors in this narrow zone, $M2$ is a more challenging test of the precision of the model in handling complex boundary phenomena.
- $M3$: *Physical consistency*: This metric assesses the model's adhesion to the governing physical laws (momentum equation), focusing on conservation and residual errors. Evaluating the momentum residual ensures consistency with the governing equations, which is essential to validate the physical plausibility of the predictions of the model.

The errors presented are calculated per pixel and normalized to the range of 0 to 100 using the following score-based equation:

$$\text{score} = 100 \times \left(1 - \frac{\log(\text{MSE}) - \log(\text{MSE}_{\min})}{\log(\text{MSE}_{\max}) - \log(\text{MSE}_{\min})}\right)$$

where $\text{MSE}_{\max} = 1$ and $\text{MSE}_{\min} = 10^{-6}$. The choice of $\text{MSE}_{\max} = 1$ represents a scenario in which the predicted solution is essentially meaningless, corresponding to the SciML model that predicts zero everywhere. Conversely, $\text{MSE}_{\min} = 10^{-6}$ reflects the numerical accuracy of the CFD simulations, which were solved using methods with residuals of a similar order. These bounds ensure that the normalized error metric aligns with both the worst-case scenario and the expected precision of the ground truth, providing an intuitive and meaningful scale for model evaluation.

First, we assess the impact of geometric representations—Signed Distance Field (SDF) and binary mask—on the models' ability to capture intricate flow attributes near the geometry. Next, we evaluate data sufficiency by varying the size of the training dataset, providing insights into each model's efficiency in learning the flow operator in data-limited regimes. This is particularly important in scientific machine learning, where data is often difficult and computationally expensive. Lastly, we examine the ability of the models to extrapolate to out-of-distribution test datasets. Specifically, we evaluate how well the models predict field solutions for problems with Reynolds numbers that are either larger or smaller than those encountered during training. In practice, scientific machine learning that can generalize to unseen data is needed for real scientific applications. This multifaceted set of experiments aims to address questions relevant to various partial differential equations (PDEs), extending beyond the lid-driven cavity problem.

## Validation dataset and hyperparameter tuning

We first divided the full dataset into 80% for model development and 20% held out for final testing. In order to monitor model generalization and guide hyperparameter tuning, we then split the 80% development set into 80% for training and 20% for validation. Model performance on the validation set is evaluated using the $M1$ global accuracy metric, which corresponds to our aggregated loss metric computed over the entire output field. Hyperparameters-including learning rate, network depth, and other architecture-specific parameters-were tuned by minimizing the validation loss over 200 epochs. Figure S.6 in the Supplementary shows example training and validation loss curves for representative models.

## Data availability

This study utilizes the FlowBench 2D Lid-Driven Cavity (LDC) dataset, which is publicly accessible on HuggingFace at https://huggingface.co/datasets/BGLab/FlowBench/tree/main/LDC_NS_2D/512x512. The dataset is licensed under a CC-BY-NC-4.0 license and serves as a benchmark for the development and evaluation of scientific machine learning (SciML) models. The data on Hugging Face is divided into three geometry sets-nurbs, harmonics, and skeleton-each comprising 1,000 samples. For this study, these sets were combined to create a total dataset of 3,000 samples.

## Code availability

The code for the models, along with training procedures and visualization scripts, is available at https://github.com/baskargroup/GeometryMatters.

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

## Acknowledgements

We gratefully acknowledge support from the NAIRR pilot program for computational access. We also acknowledge computational resources from TACC Frontera. This work is supported by the AI Research Institutes program supported by NSF and USDA-NIFA under AI Institute for Resilient Agriculture, Award No. 2021-67021-35329. We also acknowledge partial support through NSF awards CMMI-2053760 and DMREF-2323716

## Author contributions

A.R.: Data curation, Formal Analysis, Investigation, Methodology, Software, Validation, Visualization, Writing—original draft, Writing—review & editing; E.H.: Data curation, Formal Analysis, Investigation, Methodology, Software, Validation, Visualization, Writing—original draft, Writing—review & editing; A.B.: Conceptualization, Project administration, Supervision, Writing—review & editing; S.S.: Conceptualization, Project administration, Supervision, Writing—review & editing; C.H.: Conceptualization, Project administration, Supervision, Writing—review & editing; A.K.: Conceptualization, Funding acquisition, Project administration, Resources, Supervision, Writing—original draft, Writing—review & editing; B.G.: Conceptualization, Funding acquisition, Project administration, Resources, Supervision, Writing—original draft, Writing—review & editing.

## Competing interests

The authors declare no competing interests.
