## [Transparent Peer Review file · Communications Engineering]

Geometry Matters: Benchmarking Scientific Machine-Learning (SciML) Approaches for Flow Prediction Around Complex Geometries

Corresponding Author: Dr Baskar Ganapathysubramanian

Version 0:

Reviewer comments:

Reviewer #1

(Remarks to the Author)

The paper presents a comprehensive benchmarking study evaluating diverse Scientific Machine Learning (SciML) models, particularly neural operators and vision transformer-based foundation architectures, in the prediction of fluid dynamics around complex geometrical configurations. The authors identify and address an existing research gap, as previous SciML investigations into fluid dynamics have mostly concentrated on simplified geometries, leaving more realistic, complicated scenarios largely unexplored.

A key contribution of the study lies in the introduction of a unified evaluation scoring framework that integrates aspects of global accuracy, boundary layer detail, and adherence to physical consistency. This metric allows for clear, comprehensive, and rigorous comparisons between different modeling approaches. The authors further conduct a thorough, systematic evaluation comparing newer foundation models to conventional neural operators, providing clear evidence regarding model performance tendencies.

Additionally, the study offers essential practical guidance on geometric representations, namely Signed Distance Fields (SDF) and binary masks. The results illustrate that SDF geometrical representations generally offer enhanced predictive abilities compared to binary masks, provided sufficient training data are available.

The paper could be published subject to the following revisions:

1) There is recent work on the application of Transformers models in fluid dynamics, which should be cited: Generalizability of transformer-based deep learning for multidimensional turbulent flow data. *Physics of Fluids*, 2024; 36 (2): 026102; Informers for turbulent time series data forecast. *Physics of Fluids*, 2025; 37 (1): 015112; *Physics of Fluids* 2024; 36 (6): 065113. Self-supervised transformers for turbulent flow time series. *Physics of Fluids* 1 June 2024; 36 (6): 065113.

2) The authors acknowledge lingering challenges that future research must aim to overcome. Specifically, all models assessed demonstrated significant limitations in coping effectively with out-of-distribution generalization scenarios. Improving model generalization to previously unseen and diverse geometries remains a central and unresolved issue. The authors should expand the discussion on this subject.

Reviewer #3

(Remarks to the Author)

This work present an interesting comprehensive study comparing different existing neural operator and deep learning architectures for fluid flow modeling. The work specifically focuses on the effect of geometry and compares two different approaches to account for the geometry. This detailed study is expected to be very useful for the scientific machine learning community and can guide model selection in future studies. Despite my enthusiasm for this work, there are also several issues with the paper that need to be addressed. Please see comments below.

Major comments:

1- The Introduction should have better motivated the missing gap. For example, a more detailed review of prior work on

incorporating geometry in SciML as well as OOD generalization in SciML and operator learning could better motivate the present study's contribution.

2- A key issue with the comparisons is that the number of parameters in each model is not the same. Without ensuring that the number of trainable parameters are similar, it might not be very fair to compare the different methods. The number of parameters for each model needs to be reported in the main text (not Appendix) and another comparison also presented where the architectures are designed to have similar number of parameters. This was the most critical limitation of the paper that needs to be addressed.

3- The authors did not explain the validation data set. A section needs to be added to exactly explain how these were tuned. What metric was used in tuning the hyperparameters. Normally, a validation set is needed. Figure A.6. shows validation error but nothing was mentioned about the validation dataset and the text only explains training/test data split.

4- Figure 4 demonstrates that in some cases increasing the training size, makes the test error worse. Perhaps this is due to the hyperparameters not being tuned appropriately for each model? Please investigate.

5- The conclusion box in Sec 2.3 mentions that Poseidon and scOT perform better on OOD data. However, figure 6 shows that DeepONet has the best extrapolation score. Please clarify.

6- The fact that DeepONet has the best OOD performance is interesting. The authors mention that DeepONet had the smallest model size. Could this be related to the relative level of interpretability? Recent work on deep learning in physics-based problems has shown that relatively more interpretable operator learning architectures perform better on OOD data. A discussion related to this would be interesting.

7- The Abstract makes it sound like the SDF approach can notably improve the results. However, most of the results in the paper show an incremental advantage for SDF. This could be very confusing/misleading for the readers who do not study the results in detail. Please in an objective manner every place in the entire paper that SDF is mentioned to be superior to the binary approach specifically mention what percentage improvement in the score was achieved (e.g., SDF improved the score by 5% compared to the binary approach)

Minor comments:

1- The captions of tables/figures should provide more information. For example, clarify the errors are test errors. Specify what is M1, M2, etc.

2- In Sec 2.2, when the training data size is changed, please clarify if the same test is used for all or the test data for error calculation is also changed.

3- It is claimed that "neural operators like DeepONet are specifically tailored to model physical behaviors, utilizing basis functions to approximate continuous spatial fields. This inherent alignment with the governing equations...". These sentences are not very accurate. Neural operators like DeepONet are designed to work on mapping between functions $f(x)$, that is accounting for coordinates x . There is nothing in the architecture that is specific to physical systems besides considering data as a function (which all it means is that the coordinates are also considered in the mapping).

4- In the Appendix figures where contour plots are shown, please specify the score next to each one so the readers get a better feeling for the score levels.

5- The residual plots in the Appendix are a bit confusing with negative signs. Please use the absolute of the residuals and update the plots (red should represent higher residuals).

Version 1:

Reviewer comments:

Reviewer #1

(Remarks to the Author)

All the points have been addressed. The paper can be published as is.

Reviewer #3

(Remarks to the Author)

My comments were appropriately addressed. Just one minor edit:

In the revised paper, at the end of the Discussion section the authors suggest using wall shear stress solenoidality to improve robustness. Even in incompressible flows, wall shear stress is not a divergence free vector field (see work on wall shear stress fixed points and topology where wall shear stress divergence is related to near-wall normal flow and WSS fixed points.) Therefore, this statement is not true for WSS.

Response to Reviewers

Geometry Matters: Benchmarking Scientific Machine-Learning (SciML) Approaches for Flow Prediction Around Complex Geometries

We thank the reviewers for taking the time to provide insightful feedback, which improved the focus and quality of the paper. We have significantly revised the paper and have marked the changes in **magenta** in the revised document. The main revisions to the paper include:

1. We expanded the introduction to include additional, recent references on transformer-based approaches in fluid dynamics and detailed the challenges of out-of-distribution generalization in SciML.
2. We now report model parameter counts and computational performance metrics in the main text by adding a new subsection titled “Computational Performance and Parameter Analysis” in the Results section, with the corresponding table moved from the appendix, and additional plot of scores versus model size.
3. We added a new subsection titled “Validation Dataset and Hyperparameter Tuning” in the Methods section to clearly explain our validation protocol, including the use of a fixed 20% validation set and the global accuracy metric ($M1$) for tuning hyperparameters. We will also added the link to our open-source github repo of all models and parameters used in this paper.
4. We revised our discussion of training data size effects to address observed fluctuations (e.g., a slight dip for the CNO model). We added a new paragraph in the discussion on “Data Scaling” explaining that fixed hyperparameters across data regimes and stochastic training effects can account for these minor degradations without contradicting the overall trend.

We hope that with these changes, we have addressed all the major reviewer comments. We have addressed the individual concerns of the reviewers in detail below.

Reviewer #1 (Remarks to the Author):

The paper presents a comprehensive benchmarking study evaluating diverse Scientific Machine Learning (SciML) models, particularly neural operators and vision transformer-based foundation architectures, in the prediction of fluid dynamics around complex geometrical configurations. The authors identify and address an existing research gap, as previous SciML investigations into fluid dynamics have mostly concentrated on simplified geometries, leaving more realistic, complicated scenarios largely unexplored.

A key contribution of the study lies in the introduction of a unified evaluation scoring framework that integrates aspects of global accuracy, boundary layer detail, and adherence to physical consistency. This metric allows for clear, comprehensive, and rigorous comparisons between different modeling approaches. The authors further conduct a thorough, systematic evaluation comparing newer foundation models to conventional neural operators, providing clear evidence regarding model performance tendencies.

Additionally, the study offers essential practical guidance on geometric representations, namely Signed Distance Fields (SDF) and binary masks. The results illustrate that SDF geometrical representations generally offer enhanced predictive abilities compared to binary masks, provided sufficient training data are available.

Response: We thank the reviewer for an accurate and succinct summary of the contributions of our work.

The paper could be published subject to the following revisions: 1) There is recent work on the application of Transformers models in fluid dynamics, which should be cited:

Generalizability of transformer-based deep learning for multidimensional turbulent flow data. *Physics of Fluids*, 2024; 36 (2): 026102.

Informers for turbulent time series data forecast. *Physics of Fluids*, 2025; 37 (1): 015112; *Physics of Fluids* 2024; 36 (6): 065113.

Self-supervised transformers for turbulent flow time series. *Physics of Fluids* 1 June 2024; 36 (6): 065113.

Response: We have now expanded our introduction by incorporating three recent transformer-based studies in fluid dynamics showing their efficiency in capturing flow features and improves generalization for turbulent flow conditions.

2) The authors acknowledge lingering challenges that future research must aim to overcome. Specifically, all models assessed demonstrated significant limitations in coping effectively with out-of-distribution generalization scenarios. Improving model generalization to previously unseen and diverse geometries remains a central and unresolved issue. The authors should expand the discussion on this subject.

Response: We have added a detailed discussion on the limited generalization and importance and challenges of using regularizers such as physics constraints to the open questions part of the discussion.

Reviewer #3 (Remarks to the Author):

This work present an interesting comprehensive study comparing different existing neural operator and deep learning architectures for fluid flow modeling. The work specifically focuses on the effect of geometry and compares two different approaches to account for the geometry. This detailed study is expected to be very useful for the scientific machine learning community and can guide model selection in future studies. Despite my enthusiasm for this work, there are also several issues with the paper that need to be addressed. Please see comments below.

Major comments:

1). The Introduction should have better motivated the missing gap. For example, a more detailed review of prior work on incorporating geometry in SciML as well as OOD generalization in SciML and operator learning could better motivate the present study's contribution.

Response: We have now added additional references in the introduction that discuss incorporating geometry in SciML and address OOD generalization in operator learning. These additions provide a detailed review of prior work on challenges such as complex boundary conditions, arbitrary domains, and the capacity to extrapolate to unseen flow conditions, thereby better motivating the present study's contributions.

2). A key issue with the comparisons is that the number of parameters in each model is not the same. Without ensuring that the number of trainable parameters are similar, it might not be very fair to compare the different methods. The number of parameters for each model needs to be reported in the main text (not Appendix) and another comparison also presented where the architectures are designed to have similar number of parameters. This was the most critical limitation of the paper that needs to be addressed.

Response: We appreciate the reviewer raising the critical point regarding parameter counts, which we now report directly in the main text for clarity. While parameter count is a relevant metric, we believe that enforcing equal parameter count across the diverse architectures benchmarked—ranging from neural operators like FNO and DeepONet to vision transformers like scOT and Poseidon—is impractical and could obscure meaningful insights. These model families possess fundamentally distinct structural paradigms and inductive biases; for instance, the mechanisms governing information propagation in neural operators differ significantly from the attention mechanisms in transformers. Specifically, models like FNO learn global mappings in the Fourier domain, CNO leverages upsampled spatial convolutions for function spaces, while attention-based models like scOT/Poseidon inherently require large parameter counts for their query-key-value projections. These inherent architectural differences make direct parameter count comparisons less meaningful than evaluating how well each model performs according to its own design principles. As theoretical work suggests (for instance Wang and Wu²), architectural choices inherently operate at different parameter scales. Forcing equal parameter size would necessitate using non-standard configurations, potentially hurting models' ability to learn relevant features. Furthermore, studies on model scaling demonstrate that performance depends on the interplay between architecture and parameter count, with weaker inductive biases requiring more data and not parameters Bachmann et al.¹. Therefore, comparing well-tuned instances of each architecture, as is standard practice in benchmarking deep learning models, provides a more informative and realistic assessment of their strengths and weaknesses in SciML applications. We now

have added a new subsection to our results in the main paper on computational performance and parameter analysis, and we moved the table of parameter count and training/inference time to this subsection in the main paper rather than the appendix.

3). The authors did not explain the validation data set. A section needs to be added to exactly explain how these were tuned. What metric was used in tuning the hyperparameters. Normally, a validation set is needed. Figure A.6. shows validation error but nothing was mentioned about the validation dataset and the text only explains training/test data split.

Response: We have introduced a new subsection, “Validation Dataset and Hyperparameter Tuning,” in the Methods section to describe our data partitioning and tuning procedure. We begin by splitting the full dataset into 80% for training and 20% for testing, and then subdivide the 80% training portion into 80% for model training and 20% for validation, reshuffling the training subset at the start of each epoch. Validation performance is measured using the global accuracy metric, $M1$, which aggregates the loss over the entire output field, and hyperparameters are selected by minimizing this validation loss over 200 training epochs.

4). Figure 4 demonstrates that in some cases increasing the training size, makes the test error worse. Perhaps this is due to the hyperparameters not being tuned appropriately for each model? Please investigate.

Response: We note that while our overall trend shows improved performance with increased training data, minor fluctuations are expected in such complex data. Indeed, CNO was the only model to show a slight dip in test performance, particularly between 800 and 1200 samples using the binary mask representation. We confirm that all models were tuned using a validation set, but we maintained fixed hyperparameters for each model across all training sizes to isolate the effect of data scaling. Slight degradations like this can arise from stochastic training effects — including variations in random weight initialization and sensitivity to fixed learning schedules in intermediate data regimes, which may not be fully optimized for every training size. However, this fluctuation does not contradict the overall observed trend of improved performance or saturation with larger datasets.

5). The conclusion box in Sec 2.3 mentions that Poseidon and scOT perform better on OOD data. However, figure 6 shows that DeepONet has the best extrapolation score. Please clarify.

Response: Thank you for catching this error. We have revised the key takeaway box in Section 2.3 to state that all models substantially underperform when generalizing to out-of-distribution predictions, with only minimal differences between them. Although Figure 6 may show that DeepONet achieves a marginally higher extrapolation score, our overall observations indicate that no single model outperforms the others in these challenging regimes. This updated conclusion more accurately reflects our findings regarding the difficulty in predicting flow behavior at extreme Reynolds numbers.

6). The fact that DeepONet has the best OOD performance is interesting. The authors mention that DeepONet had the smallest model size. Could this be related to the relative level of interpretability? Recent work on deep learning in physics-based problems has shown that relatively more interpretable operator learning architectures perform better on OOD data. A discussion related to this would be interesting.

Response: This is a very interesting suggestion by the reviewer. We now plot parameter count vs accuracy ($M1$) as well as parameter count vs residual ($M3$) to empirically explore this. As the reviewer suggests, the lower parameter count models have marginally poorer $M1$ accuracy, but show distinctly better residual errors, suggesting an ability to learn the underlying operator better. This may potentially result in better extrapolatory performance. However, we note that DeepONet’s out-of-distribution performance is only marginally better—approximately 2–3 points higher in overall accuracy compared to the other models. We believe that the overall limited generalization ability across all models stems primarily from the purely data-driven nature of these approaches. Incorporating physics-based constraints, such as enforcing wall shear stress consistency and velocity solenoidality, may provide more regularization that could enhance the extrapolatory performance of the SciML models. We have added a discussion on this topic, as well as the challenge of balancing data fitting with physics constraints, to the open questions in the conclusion of the revised paper.

7). The Abstract makes it sound like the SDF approach can notably improve the results. However, most of the results in the paper show an incremental advantage for SDF. This could be very confusing/misleading for the readers who do not study the results in detail. Please in an objective manner every place in the entire paper that SDF is mentioned to be superior to the binary approach specifically mention what percentage improvement in the score was achieved (e.g., SDF improved the score by 5% compared to the binary approach)

Response: We have updated our abstract in the revised manuscript by explicitly stating the quantitative improvements—specifically, for vision transformer-based foundation models, binary masks yield up to a 10% improvement, while for neural operators, SDF representations provide up to a 7% improvement in the global accuracy metric $M1$. This clarification ensures that the incremental advantages are objectively communicated throughout the paper.

Minor comments:

1). The captions of tables/figures should provide more information. For example, clarify the errors are test errors. Specify what is $M1$, $M2$, etc.

Response: We have rephrased the captions of tables and figures to include more detailed information—clarifying that all error metrics represent test errors and providing definitions for the metrics $M1$ (global accuracy), $M2$ (boundary layer accuracy), and $M3$ (physical consistency).

2). In Sec 2.2, when the training data size is changed, please clarify if the same test is used for all or the test data for error calculation is also changed.

Response: In Section 2.2, we clarify that the same fixed test dataset of 600 samples is used for all training dataset size experiments.

3). It is claimed that “neural operators like DeepONet are specifically tailored to model physical behaviors, utilizing basis functions to approximate continuous spatial fields. This inherent alignment with the governing equations...”. These sentences are not very accurate. Neural operators like DeepONet are designed to work on mapping between functions $f(x)$, that is accounting for coordinates x . There is nothing in the architecture that is specific to physical systems besides considering data as a function (which all it means is that the coordinates are also considered in the mapping).

Response: We have rephrased our discussion on neural operators producing smoother and more continuous results compared to vision transformers which have more complex global dependencies and can lead to less smooth outputs.

4). In the Appendix figures where contour plots are shown, please specify the score next to each one so the readers get a better feeling for the score levels.

Response: In the Appendix field predictions figures showing contour plots, we have added the global accuracy score ($M3$) next to each log-scale error plot to help readers better gauge the score levels.

5). The residual plots in the Appendix are a bit confusing with negative signs. Please use the absolute of the residuals and update the plots (red should represent higher residuals).

Response: For the residual plots in the Appendix, we plot the log of the absolute value of the residuals (the are negative values in the colorbar is due to using log-scale). Thus, red regions in our plot correspond to larger residuals.

References

- [1] Gregor Bachmann, Sotiris Anagnostidis, and Thomas Hofmann. Scaling mlps: A tale of inductive bias. *Advances in Neural Information Processing Systems*, 36:60821–60840, 2023.
- [2] Zihao Wang and Lei Wu. Theoretical analysis of the inductive biases in deep convolutional networks. *Advances in Neural Information Processing Systems*, 36:74289–74338, 2023.

Response to Reviewers

Geometry Matters: Benchmarking Scientific Machine-Learning (SciML) Approaches for Flow Prediction Around Complex Geometries

We thank the reviewers for taking the time to provide insightful feedback, which improved the focus and quality of the paper. We thank the reviewer for pointing out that the wall shear stress is not a divergence free vector field. We have now removed that statement.